# Intrauterine Adhesions and Asherman Syndrome: A Retrospective Dive into Predictive Risk Factors, Diagnosis, and Surgical Perspectives

**DOI:** 10.3390/diagnostics15080955

**Published:** 2025-04-09

**Authors:** Loredana Maria Toma, Demetra Socolov, Daniela Matei, Sorana Anton, Raluca Balan, Emil Anton, Roxana Covali, Mihaela Tirnovanu, Handra Elicona, Theodor Pantilimonescu, Razvan Socolov

**Affiliations:** 1Department of Medical Bioscience, Faculty of Bioengineering, University of Medicine and Pharmacy “Grigore T. Popa”, 700454 Iasi, Romania; loredana-toma@umfiasi.ro (L.M.T.); daniela.matei@umfiasi.ro (D.M.); 2Clinical Hospital of Obstetrics and Gynaecology “Elena Doamna”, 700398 Iasi, Romania; raluca.balan@umfiasi.ro (R.B.); emil.anton@umfiasi.ro (E.A.); razvan.socolov@umfiasi.ro (R.S.); 3Department of Mother and Child Medicine, University of Medicine and Pharmacy “Grigore T. Popa”, 700115 Iasi, Romania; demetra.socolov@umfiasi.ro (D.S.); sorana.anton@umfiasi.ro (S.A.); mihaela.tirnovanu@umfiasi.ro (M.T.); elicona.handra@d.umfiasi.ro (H.E.); 4Clinical Hospital of Obstetrics and Gynaecology “Cuza Voda”, 700038 Iasi, Romania; 5Department of Morpho-Functional Sciences I—Histology, University of Medicine and Pharmacy “Grigore T. Popa”, 700115 Iasi, Romania; 6Department of Morpho-Functional Sciences II—Physiology, University of Medicine and Pharmacy “Grigore T. Popa”, 700115 Iasi, Romania; pantilimonescu.theodor-florin@d.umfiasi.ro; 7Clinical Hospital “Dr. C.I. Parhon”, 700503 Iasi, Romania

**Keywords:** intrauterine adhesions, Asherman syndrome, intrauterine synechiae, endometrial injury, infertility

## Abstract

**Background**: Intrauterine adhesions (IUAs) or Asherman syndrome (AS) represent pathological conditions that affect the endometrium and significantly influence female fertility through a variety of mechanisms. This study aims to identify risk factors, explore pathophysiological mechanisms, diagnostic approaches, and assess how medical background influence the development of these conditions. It also seeks to associate the severity of conditions with clinical outcomes, such as fertility, miscarriages, and menstrual cycle disorders, using American Fertility Society (AFS) scoring system. **Materials and methods**: This retrospective cohort study included 134 patients aged 18 to 45, who followed hysteroscopy between 2016 and 2024 at two hospitals in Iasi, focusing on those diagnosed with IUAs (102 patients) and AS (32 patients), based on hysteroscopic approach. The exclusions were based of factors like acute uterine bleeding, intrauterine device, obesity and other severe conditions. **Results and discussions**: Women over 35 years are more likely to develop these conditions due to prior gynaecological procedures which are often associated with fertility issues. Hysteroscopy is established as the gold standard for both diagnosis and treatment, intraoperative diagnosis representing 45.6 % of cases. Amenorrhea is a primary indicator in AS patients (OR = 26.19) and dysmenorrhea as a potential marker for IUAs (OR = 2.67). Patients with IUAs and primary infertility (82.9%) typically have an AFS score 1, corresponding to improved conception rates. Those with AS and primary infertility often present an AFS score 2 (54.5%); patients with AS and secondary infertility were linked to AFS score 3 (58.8%; *p* = 0.137). Although the incidence of miscarriages is comparable between the two groups, the timing differs: IUAs are predominantly associated with first trimester losses (64.9%), whereas AS is more commonly linked to second trimester miscarriages (45.5%; *p* = 0.001). **Conclusions:** The study highlights the necessity of a personalized approach in diagnosing and treating IUAs and AS, considering factors such as age, fertility index, and disease severity. The integration of hysteroscopic techniques with individualized treatment plans based on the patient’s unique medical profile is crucial for adequate management of IUAs and AS.

## 1. Introduction

Intrauterine adhesions (IUAs), also referred to as intrauterine synechiae, encompass a spectrum of uterine pathologies that can manifest with a variety of clinical symptoms, ranging from absent menstrual bleeding to irregular cycles, hypomenorrhea, or even normal menses, often accompanied by secondary dysmenorrhea [1]. While IUAs broadly describe the formation of scar tissue within the uterus, Asherman’s syndrome (AS) is a more specific and severe form of IUAs, characterized by significant scarring that may lead to infertility, recurrent pregnancy loss, or severe menstrual irregularities. The key distinction lies in the severity and impact on fertility and reproductive outcomes, with AS often requiring more intensive clinical intervention. These adhesions are strongly associated with adverse reproductive outcomes, including implantation failure, reduced uterine and fetal blood flow, and consequently, infertility or recurrent miscarriage, all of which significantly impair a woman’s fertility potential and desire to conceive [2,3,4]. In some instances, subfertility can be directly linked to the presence of IUAs, as these adhesions may obstruct the tubal ostia, create an inadequate or unfavorable endometrial surface for embryo implantation, or result in cervical stenosis [5].

The prevalence of IUAs is highly variable in the existing literature, with most studies focusing on symptomatic women and employing different diagnostic techniques, leading to discrepancies in reported cases. Despite the clinical relevance of IUAs, their true incidence remains poorly defined and likely underestimated, largely due to a substantial proportion of cases that remain undiagnosed, particularly in patients who are asymptomatic or misdiagnosed [3]. The increased risk of IUAs formation is thought to stem from the disruption of the endometrial healing process after mechanical injury or infectious processes that compromise the endometrial lining, leading to damage of the basal layer. Most cases are associated with pregnancy-related complications, often in the context of hypoestrogenic states, following uterine trauma or infection [6].

### 1.1. Etiology of Intrauterine Adhesions

Obstetric complications and associated procedures appear to be significant risk factors in the development of IUAs, accounting for over 90% of all cases. Hooker AB et al. (2013), in their meta-analysis, concluded that women who have undergone multiple spontaneous miscarriages or repeated dilation and curettage (D&C) procedures are at an elevated risk for the development of IUAs, with a pooled prevalence of 19.1% [3]. Unfortunately, women with diagnosed and managed IUAs following recurrent D&C for miscarriage demonstrate significantly compromised reproductive outcomes compared to those without IUAs. These include lower rates of ongoing pregnancies, reduced live birth rates, and extended time to conception. Given these challenges, primary prevention is important, and whenever possible, non-surgical alternatives to D&C, such as medical management, should be considered to optimize reproductive outcomes [7,8]. Moreover, intrauterine adhesions are observed in approximately 22% of women following postpartum curettage, a procedure commonly used to remove retained placental tissue or blood clots from the uterus after delivery [9]. Pregnancy appears to exacerbate the risk of IUAs formation, with 2% of women developing IUAs following manual removal [10]. Moreover, when procedures such as curettage, exploration, or evacuation are performed two to four weeks post-delivery, there is a substantial increase in the likelihood of IUAs formation, with reported rates ranging from 29% to 37% [11,12,13].

Another study highlighted the potential association between second-trimester TOP and the development of IUAs with prevalence rates ranging from 16.2% to 21% and an observed incidence of a pathologically wide internal os. Additionally, women who underwent first-trimester surgical TOP present a 21.2% prevalence of IUAs, with 48% of these cases being classified as moderate to severe. However, despite advancements in diagnostic modalities, the relationship between different TOP procedures and the incidence of IUAs remains inconclusive [14]. Notably, TOP is considered the most common cause of AS, both in relative and absolute terms, underscoring the significance of surgical interventions in the development of IUAs and the potential long-term reproductive consequences.

The negative pressure during uterine aspiration was identified as a significant risk factor for the development of IUAs, with an odd ratio of 125.61 (95% CI: 67.35–183.87), yielding a highly statistically significant *p*-value of <0.0001 [3]. A meta-analysis further confirmed that pelvic inflammation (*p* = 0.05), negative pressure during uterine suction (*p* < 0.0001), and the duration of uterine suction (*p* < 0.00001) were pivotal risk factors in the formation of uterine cavity adhesions [3].

In patients who had different types of myomectomies, the IUA incidence was 9.3%, with 75% of cases being minimal, and no significant difference in IUA incidence between the surgical modalities. However, 87.5% of patients with IUAs had submucosal fibroids resected, compared to 58.6% without IUAs [15]. In contrast, another two studies indicated that myomectomy had a negligible effect on the incidence of IUAs (OR = 1.35, 95% CI: 0.39–4.67), with no statistically significant correlation (*p* = 0.63) [3].

Mullerian malformations are primarily classified into three distinct categories: developmental anomalies (unicornuate uterus and mullerian hypoplasia), fusion anomalies (such as didelphys bicornuate and uteri didelphys), and resorption anomalies (septate uterus) [16]. Uterine septum is the most prevalent congenital anomaly of the female reproductive tract, with a prevalence ranging from 0.2 to 2.3% in women of reproductive age. Due to the heightened risk of miscarriage in women having this condition, the likelihood of undergoing D&C is increased, consequently elevating the risk of developing. A study indicated that, within a cohort of 522 patients diagnosed with septate uterus, 165 (31.6%) developed IUAs, with a history of one or more miscarriages being associated with an elevated risk of IUAs occurrence [17]. Robert’s uterus, a rare congenital Mullerian anomaly, is defined by an oblique septum and an asymmetrical, non-communicating uterine hemi-cavity that associate hematometra and severe dysmenorrhea being the primary clinical features. Gao et al. (2022) described a rare case of a patient diagnosed with Robert’s uterus and severe IUAs, in which successful resolution was achieved through hysteroscopic septal resection and adhesiolysis, leading to the alleviation of persistent dysmenorrhea [18].

Pelvic inflammatory disease (PID) and IUAs are closely associated, with a heightened risk of developing IUAs in women with a history of severe or chronic PID [19]. In a retrospective study of 131 patients, logistic regression analysis identified multiple abortions and CD 138 positivity as significant factors contributing to the increased risk of severe IUAs. The incidence of IUAs associated with chronic endometritis escalated in accordance with IUAs’ severity, with 10.7% of mild IUA cases, 25% of moderate cases, and 64.3% of severe cases exhibiting chronic endometritis [20].

The etiology of IUAs is predominantly iatrogenic and mechanical in general. Treatment-related factors, such as curettage for miscarriage, termination of pregnancy (TOP), removal of retained products following incomplete miscarriage, manual extraction of the placenta, and excessive endometrial trauma during hysteroscopic surgeries (polypectomy, myomectomy, and uterine septum resection), significantly contribute to the development of IUAs. Additionally, abdominal uterine surgeries, including myomectomy, caesarean section (C-section), B-lynch suturing during caesarean section, and uterine artery embolization for fibroids, have also been identified as risk factors for the formation of intrauterine adhesions [5].

### 1.2. Diagnosis of Intrauterine Adhesions

Hysteroscopy remains the most precise diagnostic modality for assessing the endometrial thickness, morphology, and any abnormalities such as IUAs [21]. Ultrasound, particularly 3D transvaginal ultrasound (3D-US), is a non-invasive, convenient, and effective initial diagnostic approach for IUAs, demonstrating a sensitivity of 98.8%, specificity of 90.8%, and accuracy of 91.4% [22]. Hysterosalpingography (HSG) can be useful for detecting IUAs, although its diagnostic accuracy is limited to approximately 50%, primarily due to challenges posed by contrast medium diffusion, air bubbles, and mucus, which may lead to false positives, as well as difficulties in diagnosing severe cervical adhesions and abnormal uterine cavity shapes. Magnetic resonance imaging (MRI) plays a critical role in evaluating the relationship between the endometrium and myometrium, thus aiding in the determination of the severity of the IUAs. When combined with ultrasound, MRI enhances diagnostic precision and prognostic evaluation [23].

### 1.3. Classification Systems for Intrauterine Adhesions

It is essential to develop a comprehensive and standardized classification system for IUAs that integrate clinical manifestation, diagnostic modalities, and patients’ reproductive history [24]. Additionally, it is crucial to assess outcomes related to restauration of normal menstrual function, fertility prognosis, and postoperative results within this system. Although several classification systems have been proposed, they are not uniformly used and have not been validated though clinical studies. The most commonly used classification systems for IUAs include those propose by March et al. (1978) [4], the American Fertility Society (AFS, 1988) [25], the European Society of Hysteroscopy (ESH), and the European Society of Gynecological Endoscopy (ESGE) [26]. March et al. (1978) were the pioneers in attempting to establish a classification system for IUAs. They described and classified the adhesion during hysteroscopy, according to the extent of involvement of the uterine cavity [4]. The AFS classification system categorizes IUAs into mild, moderate, or severe grades, based on the degree of endometrial cavity obliteration, the morphological characteristics of the adhesions, and the menstrual features of the patient, as assessed through hysteroscopy or hysterosalpingography (HSG) [25]. In 2000, Nasr et al. established a comprehensive system that formulates a prognostic score by incorporating both menstrual and obstetric history with IUA findings observed during hysteroscopic evaluation, aiming to forecast postoperative outcomes [27].

Given the potential long-term reproductive consequences of IUAs and AS, careful management of risk factors, prompt diagnosis, and effective surgical interventions are essential for optimizing reproductive health outcomes in affected women [23].

This retrospective study aims to assess the predictive factors influencing the development and severity of IUAs and AS, focusing on variables such as age, infertility history, menstrual disturbances, and prior uterine procedures. Data were collected from two maternity hospitals over an eight-year period (2016–2024), with patients categorized into two groups based on the severity of their conditions. The secondary objectives include evaluating the demographic distribution of the cohort, assessing the prevalence and diagnostic significance of common symptoms, and examining the relationship between obstetric and gynecological procedures and IUAs severity. Additionally, this study will investigate the correlation between primary and secondary infertility and IUA severity using the Nasr et al. scoring system, as well as identifying predictive factors for poor clinical outcomes after hysteroscopic adhesiolysis. Overall, the study seeks to enhance the understanding of the pathophysiological mechanisms, diagnostic efficacy, and clinical management of IUAs and AS, particularly in relation to fertility and menstrual irregularities.

## 2. Materials and Methods

### 2.1. Study Population and Design

A total of 134 patients, aged 18 to 45 years, who underwent hysteroscopy between 2016 and 2024 at the Clinical Hospital of Obstetrics and Gynecology “Elena Doamna” Iasi and the Clinical Hospital of Obstetrics and Gynecology “Cuza Voda” Iasi, were retrospectively reviewed for inclusion in this cohort study. For this retrospective cohort study, we included patients aged 18 to 45 years, as the study focuses on women of reproductive age, diagnosed with IUAs or AS based on hysteroscopic evaluation. The study also included patients presenting with clinical symptoms suggestive of IUAS, such as menstrual irregularities or subfertility. Also, patients with obstetrics or gynecological procedures known as risk factors for IUA development, such as curettage for miscarriage, C-section, intrauterine surgery, or other treatments for fibroids. Finally, only those patients who provided informed consent for the use of their clinical data in research were included, ensuring ethical compliance and protection of patient confidentiality.

Patients who met any of the following criteria were excluded: (1) weight > 100 kg, to avoid potential complications related to hysteroscopic procedures and their diagnostic accuracy; (2) presence of an intrauterine device (IUD), which could interfere with the assessment of IUAs; (3) positive result on a pregnancy test; (4) acute uterine bleeding; (5) presence of uterovaginal prolapse or severe urinary symptoms; (6) any malignancy; (7) severe intercurrent illness, including coagulation disorders, systemic diseases, or severe cardiopathies. Additionally, patients suspected of having a symptomatic reproductive tract infection, as evidenced by clinical signs or positives cultures, were excluded.

The data were gathered from two maternities over an eight-year period (2016–2024), with patients divided into two groups according to the severity of their condition: 102 patients in the IUAs group and 32 patients in the AS group, which represents a more severe form of IUAs.

### 2.2. Statistical Analysis

The Statistical Package for the Social Sciences software (SPSS, Inc., Chicago, IL, USA, version 18.0) was used for statistical analysis. Initially, the patient’s information was organized and aggregated for interpretation. Following that, the evaluation was conducted to derive additional metrics through comparative statistical methods. For the descriptive and inferential analysis, the ANOVA test was used, along with quantitative significance test (Student’s *t*-test, F-test, Pearson correlation coefficient) and multivariate analysis methods. The Kruskal–Wallis test, a non-parametric test, was specifically utilized to assess the correlation between menstrual cycle patterns and AFS scores, as these variables are categorical and do not meet the assumptions required for parametric testing. Multiple linear regression was employed to examine the correlation between a dependent variable and a set of independent variables. The Chi-Square test and the Receiver Operating Characteristic (ROC) curve were used to assess the performance and diagnostic accuracy of the model. Binary logistic regression models the relationship between a set of independent variables (Xi, categorical) and a dichotomous dependent variable (Y), with the objective of estimating the probability of the occurrence of a specific event based on the values of the independent variables. The model was developed in an incremental manner, utilizing variables that exhibited statistically significant differences in the univariate analysis. The inclusion of each new variable was determined using the likelihood ratio (LR) test. The sample size was 134, and the data were expressed as exp (B) (odd ratio) with a 95% confidence interval (CI). The objective of employing the binary logistic regression model was to assess whether age, secondary infertility, amenorrhea, and dysmenorrhea serve as significant predictors of severe IUAs, as indicated by a Nasr score categorized as severe (score 11–22) [28,29].

### 2.3. Ethical Approval

This study was conducted in accordance with the ethical principles outlined in the Declaration of Helsinki and was approved by the respective ethics committees of the participating institutions: Clinical Hospital of Obstetrics and Gynecology “Elena Doamna” and “Cuza Voda” Iasi, Romania. Ethical approval was granted by the Institutional Review Board of Clinical Hospital of Obstetrics and Gynecology “Elena Doamna” (approval no. 389/13 January 2023) and “Cuza Voda” (approval no. 13881/15 December 2020), as well as by the Ethics Committee of “Grigore T Popa” University of Medicine and Pharmacy Iasi (approval no. 402/15 February 2024). This study’s design and data management adhered to both national and international guidelines for research involving human participants.

## 3. Results

Between 2016 and 2023, the number of surgical interventions at the two University Hospitals of Obstetrics and Gynecology in Iasi has decreased, with an estimated 2316 interventions for 2024, 91% of which had been performed by August 2024. The number of pregnancies halted in progression also shows a declining trend, with 406 expected interventions in 2024, 83% of which were completed by August. Meanwhile, hysteroscopies at the Cuza Voda Maternity in Iasi have slightly increased, with an estimated 657 procedures for 2024, 63% of which had been performed by August. The number of cases with IUAs and AS admitted between 2019 and 2023 shows an increasing trend (y = 13 + 0.2x), predicting approximately 15 cases for 2024, of which 7 were admitted by September 2024.

### 3.1. Demographic and Diagnostic Features

The age distribution among patients included in the study was consistent, with the mean values closely resembling the median (33.34 vs. 34 years), and the skewness test result of 0.012 indicated that tests for statistical significance in continuous variables could be applied (Figure 1). The average age was marginally higher in patients with AS (33.97 vs. 33.14 years; *p* = 0.444), with ages ranging from 25 to 45 years. A total of 42.5% of patients exceeded the median age, with no notable differences based on the severity of the condition (42.2% vs. 43.8%; *p* = 0.874). The evaluation of the accessibility of the studied group showed that 57.5% of the total patients were from urban regions.

Among 102 females diagnosed with IUAs, 40.2% experienced primary infertility with more than a half of this group aged between 25 and 34 years (61%), while other 52% presented with secondary infertility. In the latter cohort of 32 patients diagnosed with AS, 34.4% had primary infertility, with 45.5% of these patients aged over 35 years and 56.9% associated secondary infertility (*p* = 0.831). 

Before surgery, a suggestive medical background of possible underlying conditions such as menstrual cycle disorders, infertility, a record of intrauterine procedures, or surgeries was encountered in 59 patients (44%) with a significantly higher prevalence in the group with AS (84.4% vs. 31.4%, *p* = 0.001). The medical history prior to hysteroscopic surgery revealed 32 patients with suspected IUAs, of which 56.3% were aged over 35 years (*p* = 0.036), and 65.6% were from urban areas (*p* = 0.123). In the cohort of 27 patients with suspected AS, 51.9% were between the ages of 25 and 34 years (*p* = 0.014), and 51.9% were from urban regions (*p* = 0.280).

Of the total 134 individuals included in the study, only four patients sought medical attention for suspected IUAs following hysterosalpingography (HSG) conducted to assess a potential cause of infertility. The diagnosis of IUAs was confirmed in all four cases during operative hysteroscopy (3.9% of IUAs vs. 0% of AS, *p* = 0.136). While all patients had undergone transvaginal ultrasound (TVU) before the HSG, it is noteworthy that, in these four cases, transvaginal ultrasound did not reveal the presence of intrauterine adhesions. Instead, IUAs were identified through HSG, a diagnostic method primarily used to assess tubal patency during infertility investigations. Transvaginal ultrasound assessment, using bidimensional or three-dimensional ultrasonography (2D-US, 3D-US), raised diagnostic suspicion of IUAs or AS in 70 patients from our group, accounting for 52.2% of the total study cases, and was significantly more frequent in patients diagnosed with AS (96.9% vs. 38.2%, *p* = 0.001). Intraoperative diagnosis, following hysteroscopy performed for another condition such as infertility, was established in 60 patients, representing 45.6% of the total study group, with an increased frequency in the group with IUAs (58.8% vs. 3.1%, *p* = 0.001).

### 3.2. Symptomatology of the Menstrual Cycle

Amenorrhea was identified with greater frequency in patients diagnosed with AS compared with those with IUAs (34.4% vs. 2%; *p* = 0.001), with the odd ratio (OR) for occurrence in the former group being 26.19 times higher. Hypomenorrhea was more prevalent among patients with AS although the difference did not reach statistical significance (40.6% vs. 24.5%; *p* = 0.079). Conversely, menometrorrhagia was more frequently observed in the IUAs cohort. Secondary dysmenorrhea was notably more common in patients with IUAs, with a significant OR of 2.67. Furthermore, normal regular menstrual cycles were observed more frequently in IUA patients (59.8% vs. 15.6%; *p* = 0.001), with an OR exceeding 8 times that of patients with AS (OR = 8.03; IC95%: 2.86–22.57; *p* = 0.001). These findings underscore the distinct patterns of menstrual dysfunction observed in these two patient populations (Table 1).

Amenorrhea was more frequently observed in patients with AS associated with primary infertility (45.5% vs. 33.3%, *p* = 0.211). Hypomenorrhea was more common in patients with secondary infertility, both in those with IUAs (41.5% vs. 7.3; *p* = 0.001) and those with AS (44.4% vs. 18.2%; *p* = 0.018). Dysmenorrhea was more prevalent in patients with AS and primary infertility (54.5% vs. 38.9%; *p* = 0.158). Normal menstrual cycles were more commonly found in patients with primary infertility. The best predictors for IUAs diagnostic presumption were amenorrhea (AUC = 0.662; 95% CI: 0.541–0.783; *p* = 0.006) and dysmenorrhea (AUC = 0.606; 95% CI: 0.489–0.723; *p* = 0.071). In contrast, for AS, none of these characteristics were identified as significant indicators for diagnosis.

## 4. Risk Factors for the Development and Severity of Intrauterine Adhesions

### 4.1. Obstetrical History and Mechanical Factors

Spontaneous miscarriages were identified in 53 patients, representing 39.6% of all patients included in this study. Hence, comparative analysis of the study groups revealed no significant percentage differences in the number of spontaneous miscarriages. The two populations studied had similar frequencies of one spontaneous miscarriage (AIUs 26.5% vs. AS 18.9%) and two spontaneous miscarriages (AIUs 10.8% vs. AS 15.6%). It is worth noting that about 3% of patients had a history of three spontaneous miscarriages (*p* = 0.775). Patients with IUAs had a higher frequency of spontaneous miscarriages in the first trimester compared with the second trimester (64.9% vs. 45.5%), whilst patients with AS had a higher incidence of two spontaneous abortions in the second trimester (45.5% vs. 27%; *p* = 0.001). Of 41 patients with AIU who previously had a miscarriage, 21 of them were over the age 35 years (*p* = 0.245), and 27 patients addressed from urban areas (*p* = 0.625). Additionally, 12 patients with AS-associated miscarriages, of which 7 of them residing in urban areas (*p* = 0.358), and 6 patients being over the age of 35 years (*p* = 0.359). These observations emphasize the occurrence of miscarriages in both groups with no significant association between geographical area, age, and the incidence of miscarriage in either group. Abortion was recorded in 10 patients (7.5% of the total patients) of which 9 patients were found with AIUs, 7 of whom were aged over 35 years (*p* = 0.062), and 1 patient diagnosed with AS also aged over 35 years (*p* = 0.492). Three patients from the study cohort necessitated therapeutic abortion (2.2%), of which two individuals were diagnosed with IUAs and one with AS. Previous medical abortion was found in six patients, three patients diagnosed with IUAs over 35 years old coming from urban areas, and two patients with AS both from urban areas, with one exceeding 35 years old. One patient with AS was documented in the medical sheet with preceding stillbirth (0.7% of all cases), a 42-year-old female living in the rural area, which highlights the rare but significant complications in this clinical group (Table 2).

Uterine curettage was observed in 26.5% of IUA patients and 28.1% of AS patients, with 14.7% and, respectively, 15.6% of patients undergoing two curettages. Notably is that 5.2% of patients had a history of three uterine curettages (*p* = 0.979). Among those who underwent uterine curettage for retained products following spontaneous abortion, the procedure was more frequently performed in the first trimester (52.5% vs. 50%). In contrast, patients with AS demonstrated a higher incidence of three curettages during the second trimester (12.5% vs. 16.7%; *p* = 0.001). A total of 63 patients (47% of the total cohort) underwent uterine curettage in the context of pregnancy, with 47 patients having IUAs following the procedure, of whom 23 were over 35 years old (*p* = 0.259). Additionally, 16 patients with AS underwent the procedure with only 7 over 35 years (*p* = 0.221). Postpartum curettage for retained products was performed in six patients (4.5% of the cohort), including four with AIUs and two with AS (Table 2). The majority of them were from urban areas, with heterogenous age. Caesarean section was performed in 12.7% of patients with IUAs, compared with 9.4% in the AS group (*p* = 0.599) (Table 3).

### 4.2. Gynecological History and Mechanical Factors

Dilation and curettage (D&C) for endometrial biopsy was performed in only two patients (1.5% of the total cases), both diagnosed with AS (6.3%; *p* = 0.016) and primary infertility (*p* = 0.104), over 35 years old (*p* = 0.176), and from urban areas.

The surgical treatment of fibroids varies among the studied individuals, and include myomectomy performed though different approaches such as laparotomy, laparoscopy, hysteroscopy, or uterine artery embolization. The classical approach, by laparotomy, was recorded in four patients (3% of the total cohort), two with IUAs, and two with AS and primary infertility. Another patient, aged more than 35 years old, underwent laparoscopic myomectomy, and was diagnosed with IUAs and secondary infertility. The hysteroscopic myomectomy was found in three cases (2.2% of the total cohort). Similar to these, three other patients had uterine artery embolization for uterine myomas, two of them with IUAs and one with AS, all aged over 30 years (Table 3).

Hysteroscopic interventions, such as polypectomy, septoplasty, and adhesiolysis were performed on patients with either IUAs or AS. One patient with AS and primary infertility underwent polypectomy, while septoplasty was performed on a patient with IUAs and primary infertility. Previous adhesiolysis was conducted on one patient with secondary infertility (*p* = 0.419) and two patients with primary infertility (*p* = 0.105), all aged over 35 years, with varying geographic backgrounds.

### 4.3. Pathophysiological Factors—Inflammatory/Infectious Processes

Endometritis was more frequently observed in patients with IUAs (25.4%) compared to those with AS (15.6%), though the differences were not statistically significant (*p* = 0.279). Among patients with IUAs and endometritis, 52% were aged 25–34 years, and 64% were from urban areas. In the AS group, 60% were over 35 years, with 60% from urban areas. The association between infertility type and endometritis was not statistically significant, with slightly higher frequencies in patients with secondary infertility and IUAs compared to primary infertility (24.5% vs. 22%; *p* = 0.666) as well as in those with primary infertility and AS compared to secondary infertility (27.3% vs. 11.1%; *p* = 0.318).

Unilateral or bilateral hydrosalpinx was more frequently encountered in patients with IUAs (13.7%) compared with those with AS (9.4%), although the difference was not statistically significant. Among those with hydrosalpinx, 50% of the IUAs group and all patients with AS were over 35 years old. Additionally, primary infertility was slightly more frequent in both groups, with patients having primary infertility showing a higher incidence of hydrosalpinx: IUAs group (17.1% vs. 13.2%; *p* = 0.254) and AS group (18.2% vs. 5.6%; *p* = 0.415). Unilateral or bilateral tubal obstruction without hydrosalpinx was more prevalent in patients diagnosed with AS (21.9% vs. 13.7%; *p* = 0.283), yielding an OR of 1.76, suggesting a higher likelihood of occurrence in comparison to patients with IUAs (OR = 1.76; 95% CI: 0.64–4.83; *p* = 0.283). Among those with IUAs, 51.7% were aged 25–34 years, and 57.1% were from rural areas compared with the AS group where 71.4% of patients with tubal obstruction were over 35 years old, and 66.7% were from rural areas. A history of ectopic pregnancy was more commonly reported in patients with IUAs (3.9%) compared with the AS group (3.1%) but the difference was not statistically notable (*p* = 0.833). Additionally, ectopic pregnancy was encountered in a low percentage only in patients with secondary infertility in both groups and was more common in rural areas.

### 4.4. Pathophysiological Factors—Congenital Uterine Malformations

Congenital Mullerian malformations were associated with IUAs development. In our study, this pathology was exclusively identified in patients with IUAs (10.8% vs. 0%; *p* = 0.012), exhibiting a considerably higher prevalence compared to those with AS (OR = 1.35). The majority of patients were young, with ages between 25 and 34 (81.8%), and resided in urban areas (63.6%). The prevalence of the primary infertility in this group was slightly increased among patients with IUAs, compared with secondary infertility without statistical relevance (12.2% vs. 11.3%).

### 4.5. Idiopathic Factors

Unknown factors for developing IUAs or AS were quite common in our cohort of study. Idiopathic factors were found most frequently in patients with IUAs compared with the AS group (21.6% vs. 9.4%) and had a higher OR compared to those with Asherman syndrome (OR = 2.66). The usual age group encountered in the IUAs cohort was between 25 and 34 years, and most of the patients resided in urban areas. Moreover, this study group also associated primary infertility (46.3% vs. 1.9%). In contrast, all the patients within the AS group were corelated with primary infertility (27.3% vs. 0%; *p* = 0.03).

### 4.6. Intraoperative Features

#### 4.6.1. The American Fertility Society Scoring System

In patients with IUAs, the AFS staging predominantly showed an AFS score of 1 (54.9%), whereas those with Asherman syndrome were more frequently classified as AFS score 3 (58.1%; *p* = 0.001). The correlation between the infertility type and the AFS score indicated that, for IUAs and primary infertility, AFS score 1 was predominant (82.9%), while secondary infertility was more associated with AFS score 2 (62.3%; *p* = 0.001). In contrast, in the AS group, primary infertility was linked to AFS score 2 (54.5%), while secondary infertility aligned with AFS score 3 in 58.8% of cases (*p* = 0.137). In addition, AFS score was correlated with the menstrual cycle disorders and it was found that, in patients with IUAs, amenorrhea and menometrorrhagia were most commonly linked to AFS score 3, while secondary dysmenorrhea and hypomenorrhea were associated with AFS score 2. Normal menstrual cycle instead was correlated with an AFS score 1 in 94.6% of cases (*p* = 0.001). On the contrary, in the AS group, amenorrhea, hypomenorrhea, and dysmenorrhea were mainly associated with AFS score 3 (50%, 50%, and 44.4%), while menometrorrhagia was correlated with AFS score 2 (23.1 %). A normal menstrual cycle was instead more commonly assigned with AFS score 2 (*p* = 0.002) (Table 4).

The analysis of the correlation between conception prognosis and viable pregnancy revealed significant findings. Among patients with IUAs, the prognosis varied according to the AFS classification. Specifically, 71.4% of patients with AFS score 1 had an excellent prognosis, exceeding 75%, 68.3% of patients with AFS score 2 had a prognosis ranging between 50% and 75%, and 75% of patients with AFS score 3 had a prognosis between 25% and 50% (*p* = 0.001).

#### 4.6.2. Predicting Outcomes After Hysteroscopic Adhesiolysis Using the Nasr et al. (2000) [27] Scoring System

In patients presenting with IUAs, the Nasr classification predominantly indicated a mild score (54.9%), which was associated with favorable prognosis and positive outcomes following treatment. Conversely, in individuals diagnosed with AS, a more severe Nasr score (11–22) was observed (78%, *p* = 0.001), which demonstrated statistical significance in predicting a poorer prognosis (Figure 2).

The relationship between infertility and prognosis after hysteroscopic adhesiolysis, as classified according to the Nasr score, revealed notable patterns. In patients with IUAs and primary infertility, the most prevalent Nasr score was 1–4 (87.8%), while in patients with intrauterine adhesions and secondary infertility, the predominant Nasr score was 5–10 (63%, *p* = 0.001), indicating statistical significance. In contrast, within the cohort of patients diagnosed with AS, the Nasr score was predominantly 11–22 for both primary infertility (54.5%) and secondary infertility (94.4%, *p* = 0.013), showing a poor outcome for these patients after the treatment, further underscoring the statistical significance of the findings.

The correlation between menstrual cycle symptoms and the Nasr scoring system demonstrated noteworthy associations in patients with IUAs: 10% of the patients with amenorrhea showed a severe Nasr score, which is statistically significant (*p* = 0.042), while a moderate Nasr score was found predominantly within the IUAs cohort with hypomenorrhea, (90% of patients, *p* = 0.001), demonstrating strong statistical significance. Moreover, 85.7% of patients with IUAs and normal menstrual cycle presented a mild Nasr score (*p* = 0.001). A moderate Nasr score (5–10) was observed in 19.4% of cases with IUAs associated with menometrorrhagia (*p* = 0.601) and in 33.3% of cases with IUAs and secondary dysmenorrhea (*p* = 0.601); however, neither of these associations was statistically significant (Table 5).

As a result of the binary logistic regression analysis conducted on patients with IUAs and using the Nasr scoring system, four models were developed utilizing four statistically significant variables (Table 6).

Model 1 indicates that the likelihood of severe IUA is more than twice as high in the 25–34 age group (OR = 2.33; 95% CI: 1.90–6.07; *p* = 0.042). The subsequent models, developed in an incremental manner, highlight significant associations. Model 2 indicates that the probability of severe IUAs is more than 4.3 times higher in patients with secondary infertility (OR = 4.36; 95% CI: 1.49–8.97; *p* = 0.001). Model 3 illustrates that the probability of severe IUAs is more than 8 times higher in the presence of amenorrhea (OR = 8.64; 95% CI: 4.80–19.78; *p* = 0.001). Model 4 shows that the probability of severe IUAs increases by approximately 8.6 times when dysmenorrhea is present (OR = 8.593; 95% CI: 1.89–13.42; *p* = 0.001) (Table 6). These findings suggest that age, secondary infertility, amenorrhea, and dysmenorrhea are significant predictors of severe IUA, as reflected by a Nasr score of 11–22, providing valuable insights for early identification and management of at-risk patients.

## 5. Discussions

In our study, we found no notable differences in age distribution between patients with IUAs and AS, as the mean age was closely matched between the two cohorts and comparable with the literature [30]. Additionally, the fact that age was not linked with the severity of the condition shows that life stage is not a deceiving element in how severe the pathology becomes compared with others such as past treatments or health issues, but it can influence patient’s fertility. On the other side, females over the age of 35 are at higher risk to develop IUAs or AS due to a background of various gynecological procedures, miscarriages, or genital infections throughout their lives [23].

The diagnosis of IUAs or AS is by direct visualization during hysteroscopy for the assessment of the uterine cavity. Other diagnostic methods such as hysterosalpingography or even transvaginal ultrasonography present a lower diagnostic accuracy. This explains the reduced number of patients from this cohort diagnosed following hysterosalpingography [31]. Although all patients underwent transvaginal ultrasound prior to the final diagnosis, it is noteworthy that, in some instances, ultrasound failed to detect the presence of IUAs. In contrast, the IUAs were identified through HSG in four cases, a diagnostic tool primarily utilized to assess tubal patency during infertility evaluations. One limitation of this study is that the diagnostic methods were not carried out by a small, well-selected group of 2–3 experienced physicians, particularly for ultrasound, which may have influenced the results. This is important because the cohort includes cases wherein IUAs were diagnosed through methods other than ultrasound as the initial diagnostic tool, such as hysterosalpingography, which may have resulted in some variability in diagnosis and the overall findings of the study. Transvaginal ultrasonography (TV-US), particularly the three-dimensional TV-US, has been shown to be an effective non-invasive diagnostic method, but still not as efficient as the hysteroscopic approach.

Hence, the high rate of intraoperative diagnoses of IUA cases from our study reflect these usual findings in the literature and align with the current trend when referring to 3D TV-US. The utmost adequate method for diagnosing IUAs is hysteroscopy, whilst AS may necessitate a more thorough clinical background and visual examination [23,32]. Concerning the prevalence of AS in younger age groups, our results emphasize the evidence available, according to which the younger females tend to present with AS due to uterine injury related to a pregnancy background. Also, the predominance of IUAs in females from urban areas confirm the belief that a better access to healthcare services and advanced diagnostic techniques increase the patient’s addressability, diagnostic, and management.

Our study showed that amenorrhea can be considered a significant predictor for AS and could serve as a crucial clinical marker to diagnose AS in females that have had a prior uterine trauma or surgery. This symptom is the consequence of severe endometrial damage, which impairs the uterine cavity’s ability to shed the menstrual lining [4,33]. However, it is well-established that endometrial disruption can present with a range of symptoms which varies from amenorrhea, hypomenorrhea, dysmenorrhea to irregular uterine bleeding, and infertility [34,35]. The finding that dysmenorrhea was identified as a potential diagnostic predictor for IUAs, despite lacking statistical significance, suggests that dysmenorrhea is frequently associated with functional uterine pathologies like uterine fibroids, adenomyosis, or endometriosis.

Hypomenorrhea was more common in patients with secondary infertility, suggesting that these patients present a less severe form of endometrial damage compared with those that associate primary infertility. On the other side, normal menstrual cycles were associated with primary infertility and this may reflect that endometrial damage is less important in IUAs compared with AS; the adhesion films are less extensive and do not always disrupt the menstrual flow, showing that the fertility issue can be present in the absence of menstrual cycle disorders associated with IUAs. The lack of diagnostic predictors calls for a comprehensive diagnostic approach for these two pathologies, including patient’s medical background, TV-US, and hysteroscopy [34,35].

This study highlights several notable findings regarding the history of spontaneous miscarriages, surgical procedures, and complications in patients diagnosed with IUAs and AS. It is clear that the impact of uterine trauma or intrauterine surgery affects the reproductive health and pregnancy outcomes [36]. The incidence of spontaneous abortion was comparable between IUAs and AS patients, showing the same negative impact on pregnancy outcomes by increasing the risk of miscarriage due to anatomic alteration of the uterine cavity and endometrial compromise. Intrauterine adhesions were more prevalently affiliated with early miscarriages, whilst AS was associated with an increased incidence of second-trimester miscarriages compared with first-trimester ones. It is possible that the endometrium is more prone to disruptions caused by adhesions in the first trimester, whereas patients with AS may experience more extensive and dense adhesions, potentially to reduce uterine and fetal blood flow or to determine placental insufficiency due to endometrial adhesions or fibrosis [23]. Still, the similar frequency of miscarriages in the two groups indicates that the endometrial trauma has a profound impact, regardless of the extent of the adhesions.

The lack of notable correlations between age, geographical location, and the incidence of miscarriages that were observed in this retrospective study propose that intrinsic factors such as adhesions development or endometrial fibrosis influence IUA and AS, rather than extrinsic factors such as age or location. This underlines the existing data from the literature which emphasize the underlying uterine mechanisms beyond this condition such as endometrial damage or intrauterine surgeries as main factors that contribute to spontaneous abortion risk in IUAs and AS.

Similar rates of one uterine curettage following retain products after spontaneous abortion in patients with IUAs and AS (26.5% vs. 28.1%) are concerning. A small group, representing 5.2% of patients, presents a history of three curettages (*p* = 0.979) which shows the issues of the use of uterine curettages and emphasizes the necessity for meticulous management in order to decrease further complications considering fertility [37,38,39]. In addition, the occurrence of stillbirth in one AS patient and therapeutic abortions in others points to the likelihood of severe pregnancy complications in females with AS, such as the ability to carry a pregnancy to term.

The variety of gynecologic surgical procedures in this cohort highlights the personalized treatment approach based on clinical features such as patient’s age, fertility status, and severity of the condition. In our study, only two cases (1.5%) diagnosed with AS and primary infertility identified D&C procedures as a causative factor, despite endometrial biopsy via D&C being common in our country. The low incidence observed in the two maternity centers can be attributed to the younger age of patients, where D&C for biopsy is less prevalent, particularly due to the lower incidence of oncological conditions in this group. Additionally, benign uterine pathologies, such as polyps and fibroids, are primarily diagnosed via ultrasonography, with hysteroscopic procedures being the preferred management approach. The patient selection focuses on those diagnosed preoperatively and intraoperatively with IUAs, emphasizing risk factors rather than including patients with risk factors to analyze the incidence of IUAs.

The range of multiple approaches in patients with uterine fibroids according to their clinical status is shown in this study, tailoring each patient’s specific needs and medical condition. For this pathology, the treatment varies from laparotomy approach to minimally invasive surgery such as laparoscopy or hysteroscopy, and uterine artery embolization, according to patient’s reproductive history, severity of condition, other comorbidities, and other complicating factors such as the presence of adhesions. Moreover, the diverse array of hysteroscopic techniques used to remove polyps, or fibroids, and to correct uterine cavity anatomy when IUAs, AS, or uterine septum are present, reflect the customized treatment strategy in managing these pathologies. A wider age spectrum was observed between these patients compared with those that associated pregnancy-related history, which underlines the importance of taking into account the patient’s age, fertility status, and desire, and other factors when a surgical intervention is to be planned [40,41,42,43,44].

An intriguing finding in this study is the link between congenital Mullerian malformations and the development of IUAs in contrast to the AS cohort (OR = 1.35), showing that a disturbance of uterine anatomy may predispose patients to develop IUAs [18]. The fact that the majority of the patients with congenital uterine malformation and IUAs were young, aged between 25 and 34 years, and living in urban areas, suggests that these factors may play a role in the clinical outcome and diagnosis of IUAs. Moreover, the results reveal an important prevalence of idiopathic factors among both patients with IUAs and AS. The higher incidence of unknown etiology is more prevalent in the IUAs group and it reveals the complexity of the condition’s pathogenesis, pointing to the fact that there are still some mechanisms that require further investigations. It is also worth noting that females with an idiopathic cause of IUAs more frequently experienced primary infertility, which is in contrast to the AS group. Given the young age group of IUA patients and their associations with primary infertility, there is a need for further investigation into the underlying causes of these conditions, particularly when the etiology remains unknown.

Our study underlies the considerable link between AFS scoring [25], infertility type, menstrual cycle disorders, and conception prognosis in the two groups. It was found that patients with IUAs present less severe IUAs, scoring an AFS 1, whilst patients with AS were predominantly assigned AFS score 3, showing severe damage of the endometrium. Also, primary infertility was more linked to AFS score 1, compared to secondary infertility that was more aligned with AFS score 2. On the other side, in the AS group, patients with primary infertility had an AFS score 2 and those with secondary infertility were associated with AFS score 3. Regarding the menstrual disturbances, the most severe symptoms such as amenorrhea and menometrorrhagia were mostly common in the critical stages, particularly AFS score 3, demonstrating a strong association between the severity of the disease and menstrual cycle disfunction. Regular menses were more frequently associated with the IUAs cohort, scoring an AFS 1, whilst the AS patients with normal menses were found to have and AFS score 2. The predictive value of AFS score in assessing fertility outcomes showed that IUAs patients with AFS score 1 presented the best prognosis (71% achieving more than a 75% chance to become pregnant), whereas the group with AFS score 3 had a lower prognosis, between 25 and 50 %.

Moreover, this study underscores the significant role of Nasr et al.’s (2000) scoring system in predicting clinical outcomes after hysteroscopic adhesiolysis in patients with IUAs and AS. The findings highlight that more severe Nasr scores, particularly in patients with AS, are associated with poorer prognoses, emphasizing the importance of early identification and intervention. Key factors such as secondary infertility, age (especially in the 25–34 age group), and menstrual symptoms (such as amenorrhea and dysmenorrhea) were found to significantly predict the severity of IUAs and treatment outcomes. Furthermore, it demonstrates that menstrual disturbances, like amenorrhea and hypomenorrhea, are strong indicators of severe IUAs, while a normal menstrual pattern correlates with a milder form of IUAs. The regression models developed in this study suggest that these clinical variables can serve as useful predictors for identifying patients at higher risk of poor outcomes, enabling more targeted management.

Overall, this study correlates important features about the diagnosis, clinical symptoms, intraoperative findings, and reproductive outcomes in patients with IUAs and AS. Females aging over 35 years old are more prone to develop IUAs or AS associated with fertility issues due to possible gynecological procedures they might have during life. Hysteroscopy proved to be the gold standard for diagnosis and treatment, with amenorrhea identified as a main clinical signal for patients with AS and medical background, and dysmenorrhea as a potential indicator for IUAs. The degree of endometrial damage, as assessed by the AFS score, is closely correlated with menstrual cycle disturbances and fertility outcomes, showing that a decreased AFS score is associated with increased conception outcomes. Regardless of whether the rates of spontaneous abortion are similar between the groups, the timing is distinct. The IUAs are correlated with first-trimester miscarriages, whereas AS is more prevalent following second-trimester miscarriages [23,45].

## 6. Conclusions

This study underscores the importance of a comprehensive and individualized strategy for the practitioner to establish the diagnosis and treatment, taking into consideration variables such as patient’s age, fertility status, and severity of condition when planning the surgical treatment. The use of hysteroscopic techniques along with personalized treatment strategies tailored to the unique medical profile of the patient is essential for a successful management in patients with IUAs and AS.

## Figures and Tables

**Figure 1 diagnostics-15-00955-f001:**
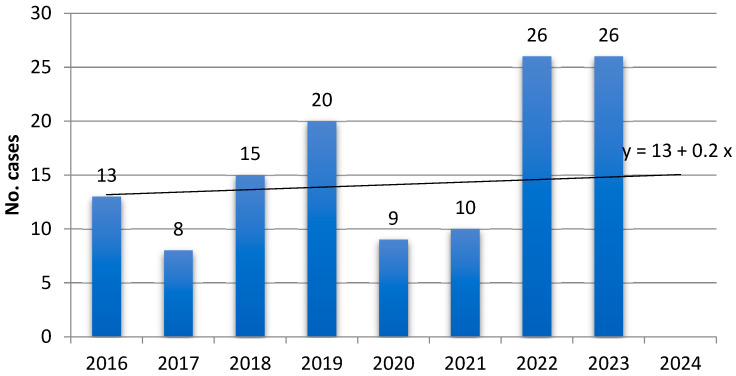
Distribution of cases with IUAs and AS hospitalized between 2016 and 2023.

**Figure 2 diagnostics-15-00955-f002:**
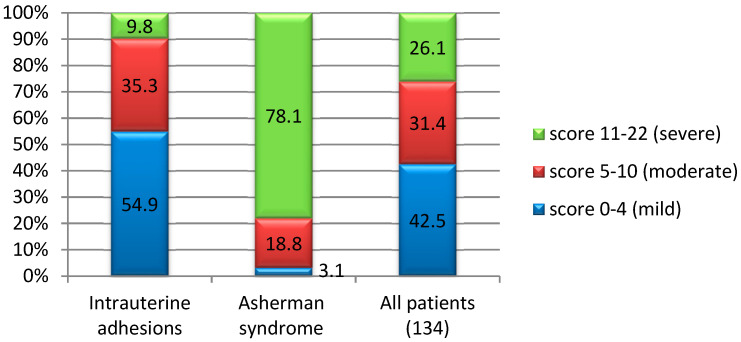
The application of the clinico-hysteroscopic scoring system of IUAs proposed by Nasr et al. (2000)—prognostic outcomes in our cohort between patients with IUAs and AS [27].

**Table 1 diagnostics-15-00955-t001:** Menstrual cycle disorders in IUAs and AS patients.

Symptomatology of the Menstrual Cycle	AS (no. 32)		IUAs (no. 102)		Chi2 Test (*p* Value)	OR	CI 95%
	no.	%	no.	%			
Amenorrhea	11	34.4	2	2.0	0.001	26.19	5.40–127.0
Hypomenorrhea	13	40.6	25	24.5	0.079	2.11	0.91–4.87
Menometrorrhagia	3	9.4	15	14.7	0.425	1.67_A_	0.45–6.17
Dysmenorrhea	14	43.8	23	22.5	0.023	2.67	1.15–6.18
Normal MC	5	15.6	61	59.8	0.001	8.03_A_	2.86–22.57

IUAs—intrauterine adhesions, AS—Asherman’s syndrome, MC—menstrual cycle, OR—odds ratio, CI—confidence interval.

**Table 2 diagnostics-15-00955-t002:** The obstetrical history among patients with intrauterine adhesions and Asherman syndrome.

Obstetrical History	Primary Infertility		Secondary Infertility		CHI2 Test
	No.	%	No.	%	
**Intrauterine adhesions**	**no = 41**		**no = 53**		***p* Value**
Spontaneous abortion (complete or incomplete)	0	0.0	41	77.4	0.001
Complete spontaneous abortion	0	0.0	37	69.8	0.001
Induced abortion	0	0.0	9	17.0	0.002
Therapeutic abortion	0	0.0	2	3.8	0.265
Medical abortion	0	0.0	4	7.5	0.068
Stillbirth	0	0.0	0	0.0	nc
Dilation and curettage	0	0.0	41	77.4	0.001
Postpartum curettage	0	0.0	4	7.5	0.068
Asherman syndrome	no = 11		no = 18		*p* value
Spontaneous abortion (complete or incomplete)	0	0.0	11	61.1	0.003
Complete spontaneous abortion	0	0.0	10	55.6	0.048
Induced abortion	0	0.0	1	5.6	0.556
Therapeutic abortion	0	0.0	1	5.6	0.556
Medical abortion	0	0.0	2	11.1	0.300
Stillbirth	0	0.0	1	5.6	0.556
Dilation and curettage	0	0.0	15	93.8	0.001
Postpartum curettage	0	0.0	2	11.1	0.300

nc = not calculated.

**Table 3 diagnostics-15-00955-t003:** Gynecological history among patients in the study groups.

Surgical Interventions	No Infertility		Primary Infertility		Secondary Infertility		Chi Square Test
	No.	%	No.	%	No.	%	
**Intrauterine Adhesions**							
No.	8		41		53		*p* value
C-section	6	75.0	-	-	7	13.2	0.001
D&C for biopsy	-	-	-	-	-	-	-
Classic myomectomy	-	-	-	-	2	3.8	0.265
Laparoscopic myomectomy	-	-	-	-	1	1.9	0.517
Hysteroscopic myomectomy	-	-	1	2.4	2	3.8	0.728
UAE	-	-	1	2.4	1	1.9	0.834
Hysteroscopic polypectomy	-	-	-	-	-	-	-
Hysteroscopic septoplasty	-	-	1	2.4	-	-	0.399
Hysteroscopic adhesiolysis	-	-	-	-	1	1.9	0.517
Asherman Syndrome							
No.	3		11		18		*p* value
C-section	1	33.3	-	-	2	11.1	0.171
D&C for biopsy	-	-	2	18.2	-	-	0.104
Classic myomectomy	-	-	2	18.2	-	-	0.104
Laparoscopic myomectomy	-	-	-	-	-	-	-
Hysteroscopic myomectomy	-	-	-	-	-	-	-
UAE	-	-	1	9.1	-	-	0.333
Hysteroscopic polypectomy	-	-	1	9.1	-	-	0.333
Hysteroscopic septoplasty	-	-	-	-	-	-	-
Hysteroscopic adhesiolysis	-	-	2	18.2	-	-	0.105

C-section—caesarean section; D&C—dilation and curettage; UAE—uterine artery embolization.

**Table 4 diagnostics-15-00955-t004:** Correlation between menstrual cycle pattern and AFS score for IUAs.

Menstrual Cycle Pattern	AFS Score 1		AFS Score 2		AFS Score 3		*p* Value
	no.	%	no.	%	no.	%	
Intrauterine adhesions							
Amenorrhea	-	-	-	-	2	50.0	0.001
Hypomenorrhea	-	-	24	57.1	1	25.0	0.001
Menometrorrhagia	3	5.4	10	23.8	2	50.0	0.006
Dysmenorrhea	9	16.1	13	31.0	1	25.0	0.125
No symptoms	53	94.6	8	19.0	-	-	0.001
Asherman syndrome							
Amenorrhea	-	-	2	15.4	9	50.0	0.040
Hypomenorrhea	-	-	3	23.1	9	50.0	0.123
Menometrorrhagia	-	-	3	23.1	-	-	0.017
Dysmenorrhea	-	-	5	38.5	8	44.4	0.739
No symptoms	-	-	5	38.5	-	-	0.002

**Table 5 diagnostics-15-00955-t005:** Correlations between menstrual cycle symptoms and the Nasr scoring system (Nasr et al., 2000) [27].

Menstrual Cycle Pattern	Score 0–4		Score 5–10		Score 11–22		*p* Value
	*n*	%	*n*	%	*n*	%	
Intrauterine adhesions							
Amenorrhea	-	-	1	2.8	1	10.0	0.042
Hypomenorrhea	1	1.8	15	41.7	9	90.0	0.001
Menometrorrhagia	7	12.5	7	19.4	4	10.0	0.601
Dysmenorrhea	10	17.9	12	33.3	1	10.0	0.135
Normal MC	48	85.7	13	36.1	-	-	0.001
Asherman syndrome							
Amenorrhea	-	-	-	-	11	44.0	0.032
Hypomenorrhea	-	-	2	33.3	11	44.0	0.523
Menometrorrhagia	-	-	-	-	3	12.0	0.457
Dysmenorrhea	1	100.0	2	33.3	1	10.0	0.038
Normal MC	1	100.0	4	66.7	-	-	0.001

**Table 6 diagnostics-15-00955-t006:** Binary logistic regression models predictors of IUAs according to severity—Independent variables: age, infertility, amenorrhea, and dysmenorrhea.

Logistic Regression Models	Independent Variables	Odds Ratio (95% CI)	*p* Value
**IUAs (Yes/No) Nasr = Severe**			
**Model 1**	Age group 25–34 y	2.333 (1.897–6.072)	0.042
**Model 2**	Age group 25–34 y	1.851 (1.616–5.565)	0.047
	Secondary infertility	4.363 (1.489–8.968)	0.001
**Model 3**	Age group 25–34 y	1.184 (1.351–3.993)	0.045
	Secondary infertility	1.730 (1.146–2.449)	0.072
	Amenorrhea	8.047 (1.827–17.294)	0.001
**Model 4**	Age group 25–34 y	1.717 (1.179–2.876)	0.039
	Secondary infertility	1.967 (1.057–16.340)	0.048
	Amenorrhea	8.638 (4.804–10.783)	0.001
	Dysmenorrhea	8.593 (1.885–13.416)	0.001

IUAs = Intrauterine adhesions; y = years old.

## Data Availability

The original contributions presented in this study are included in the article. Further inquiries can be directed to the corresponding author.

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
