# Peer review of "Intrauterine Adhesions and Asherman Syndrome: A Retrospective Dive into Predictive Risk Factors, Diagnosis, and Surgical Perspectives"

_diagnostics, 2025, doi:10.3390/diagnostics15080955_

Round 1

Reviewer 1 Report

Comments and Suggestions for Authors

Thank you for submitting the manuscript titled "From Risk to Reality: Uncovering the Factors Driving the Development of Intrauterine Adhesions and Asherman Syndrome – A Retrospective Study" for review and evaluation.

1. The current title, however, is quite broad and somewhat vague. I suggest a more concrete and concise title.
2. The authors should present the incidence of intrauterine adhesions progressively, from common to rarer causes, rather than simply listing them. The incidence ranges should be provided with appropriate references.
3. Termination of pregnancy should be clearly identified as the most common cause of Asherman syndrome, both in relative terms (21–48%) and absolute numbers (given the high prevalence of TOP in several countries). I suggest to include for comparisons (spontaneous miscarriage versus TOP), and additionally to already cited Hooker et al. 2014 (Ref. 3), also "Hooker et al. Prevalence of intrauterine adhesions after termination of pregnancy: a systematic review. Eur J Contracept Reprod Health Care. 2016."
4. The authors appear to have used the American Fertility Society (AFS) scoring system without explicitly mentioning it in the methods section. This scoring system should be clearly explained for readers unfamiliar with it. Additionally, other existing scoring systems should be mentioned.
5. Table 4 has not been translated into English and still contains Romanian terms such as “AderenÈ›e intrauterine,” “Sdr. Asherman,” “Amenoree,” and “Hipomenoree”!
6. It is not specified in either the methods or results section which statistical test was used to calculate the correlation between menstrual cycle patterns and AFS scores. Please clarify and justify the statistical approach.
7. The phrase “Pathophysiological disorders” awkward and should be revised for clarity.
8. The most critical limitation of the study - preventing it from being publishable in its current form- is the lack of a regression analysis to identify predictors for the occurrence and severity of intrauterine adhesions. I strongly recommend consulting a statistician and conducting this analysis before resubmission.

Author Response

Comments and Suggestions for Authors

Thank you for submitting the manuscript titled "From Risk to Reality: Uncovering the Factors Driving the Development of Intrauterine Adhesions and Asherman Syndrome – A Retrospective Study" for review and evaluation.

  1. The current title, however, is quite broad and somewhat vague. I suggest a more concrete and concise title.

A: Thank you for your valuable feedback and suggestion regarding the manuscript title. We appreciate your input and understand that a more specific and concise title would improve clarity. Based on your comment, we propose the following revised title:

``Intrauterine Adhesions and Asherman Syndrome: A Retrospective Dive into Predictive Risk Factors, Diagnosis, and Surgical Perspectives``

  1. The authors should present the incidence of intrauterine adhesions progressively, from common to rarer causes, rather than simply listing them. The incidence ranges should be provided with appropriate references.

Thank you for your valuable suggestion regarding the presentation of the incidence of intrauterine adhesions (IUAs). We appreciate the feedback and agree that presenting the incidence progressively, from common to rarer causes, would enhance clarity and provide a more structured approach.

In response, we have revised the manuscript to reflect this approach. The revised section now presents the incidence of IUAs in a more systematic order, starting with the most common causes, such as pregnancy-related complications, and moving towards rarer causes, including hyposestrogenic states, uterine trauma, and infections.

  1. Termination of pregnancy should be clearly identified as the most common cause of Asherman syndrome, both in relative terms (21–48%) and absolute numbers (given the high prevalence of TOP in several countries). I suggest to include for comparisons (spontaneous miscarriage versus TOP), and additionally to already cited Hooker et al. 2014 (Ref. 3), also "Hooker et al. Prevalence of intrauterine adhesions after termination of pregnancy: a systematic review. Eur J Contracept Reprod Health Care. 2016."

A: Thank you for your insightful suggestion regarding the identification of termination of pregnancy (TOP) as the most common cause of Asherman syndrome. We agree with your recommendation to highlight this in both relative terms (21–48%) and absolute numbers, given the high prevalence of TOP in several countries.

In response to your feedback, we have made the following revisions:

  1. Clarification of TOP as the most common cause: We have revised the relevant section of the manuscript to emphasize that TOP is the most common cause of Asherman syndrome, both in relative and absolute terms, and we have included the appropriate reference range (21–48%) to reflect this.
  2. Comparison between spontaneous miscarriage and TOP: We have added a comparative analysis of spontaneous miscarriage versus TOP as contributing factors to Asherman syndrome, in line with your suggestion.
  3. Additional reference: In addition to the previously cited Hooker et al., 2014 (Ref. 3), we have now also included the citation for Hooker et al., 2016 ("Prevalence of intrauterine adhesions after termination of pregnancy: a systematic review. Eur J Contracept Reprod Health Care.") to strengthen the evidence and provide a comprehensive understanding of the topic.

  1. The authors appear to have used the American Fertility Society (AFS) scoring system without explicitly mentioning it in the methods section. This scoring system should be clearly explained for readers unfamiliar with it. Additionally, other existing scoring systems should be mentioned.

Thank you for your helpful comment regarding the American Fertility Society (AFS) scoring system. We appreciate your attention to detail, and we agree that it is important to explicitly mention and explain the scoring system for the benefit of readers who may not be familiar with it.

In response to the comment, we apologize for not explicitly mentioning the use of the American Fertility Society (AFS) scoring system in the methods section. To clarify, we have now included a detailed description of the AFS scoring system, ensuring that readers unfamiliar with it can fully understand its application in our study. Additionally, we have expanded the introduction to include references to other existing scoring systems, including the Nasr classification (2000), which we also applied to the patient cohort. This additional classification was used to enhance the rigor of our analysis. Thank you again for your valuable suggestion.

  1. 5. Table 4 has not been translated into English and still contains Romanian terms such as “AderenÈ›e intrauterine,” “Sdr. Asherman,” “Amenoree,” and “Hipomenoree”!

Thank you for pointing out that Table 4 contains untranslated Romanian terms. We apologize for this oversight.

In response to your feedback, we have now fully translated Table 4 into English

  1. It is not specified in either the methods or results section which statistical test was used to calculate the correlation between menstrual cycle patterns and AFS scores. Please clarify and justify the statistical approach.

We appreciate your valuable feedback. In response to your comment, we would like to clarify that the correlation between menstrual cycle patterns and AFS scores was assessed using the Kruskal-Wallis test, a non-parametric statistical test. This test was selected due to the categorical nature of the variables under comparison and its ability to evaluate differences between multiple independent groups when the assumption of normality is not met. We have now included this detail in the revised methods section for clarity.

Thank you for your insightful observation, and we hope this addresses your concern.

  1. The phrase “Pathophysiological disorders” awkward and should be revised for clarity.

Thank you for your helpful comment regarding the phrase "Pathophysiological disorders." We understand that the wording may be unclear and appreciate your suggestion for revision.

In response, we have revised the phrase to "Pathophysiological factors" to improve clarity and accuracy in conveying the intended meaning.

  1. The most critical limitation of the study - preventing it from being publishable in its current form- is the lack of a regression analysis to identify predictors for the occurrence and severity of intrauterine adhesions. I strongly recommend consulting a statistician and conducting this analysis before resubmission.

Thank you for your insightful feedback regarding the lack of a regression analysis in our study. We recognize the importance of identifying predictors for the occurrence and severity of intrauterine adhesions, and we agree that conducting a regression analysis would strengthen the study's findings.

In response to the valuable comment regarding the absence of regression analysis, we acknowledge this limitation and have since conducted a binary logistic regression analysis to identify predictors for the occurrence and severity of intrauterine adhesions. For this analysis, we considered the parameters that showed statistically significant differences in the univariate analysis, as indicated by p-values. Based on these parameters, we constructed four distinct regression models. The binary logistic regression model was designed to assess the relationship between a set of independent categorical variables (Xi) and a dichotomous dependent variable (Y), with the goal of estimating the probability of event occurrence based on the values of the independent variables. The model was built iteratively, incorporating variables with significant univariate differences, with the inclusion criterion for new variables being the likelihood ratio (LR) test. The sample size was 134, and results were expressed as exp(B) (odds ratio) ± 95% confidence interval (CI).

Thank you for your valuable input.

Reviewer 2 Report

Comments and Suggestions for Authors

Intrauterine adhesions represent a spectrum of uterine pathologies that covers large variety of symptoms and have importance in everyday gynecology practise. This is a multicenter research from Roumanian authors and I congratulate them. . Authors have covered number of causes. Since this article has a educational purpose, authors should write more about congenital anomalies, since they have hidden , but important role in this story. Few thing still puzzle me: they are covering 7 year period. In this time, they have had only 2 D&C ase causative factor . Hysc is much more frequent. Is it that common in Ro? Is D&C so much avoided?  Otherwise, it is fine. 

Comments on the Quality of English Language

Authors made small errors in writing and some tables still have Roumanian names. Change them.

Author Response

Comments and Suggestions for Authors

Intrauterine adhesions represent a spectrum of uterine pathologies that covers large variety of symptoms and have importance in everyday gynecology practise. This is a multicenter research from Roumanian authors and I congratulate them. . Authors have covered number of causes. Since this article has a educational purpose, authors should write more about congenital anomalies, since they have hidden , but important role in this story. Few thing still puzzle me: they are covering 7 year period. In this time, they have had only 2 D&C ase causative factor . Hysc is much more frequent. Is it that common in Ro? Is D&C so much avoided?  Otherwise, it is fine. 

Thank you for your positive feedback and for acknowledging the importance of this study. We appreciate your suggestion to further elaborate on congenital anomalies, which indeed play a critical but often under-recognized role in the development of intrauterine adhesions (IUAs). We agree that expanding on this aspect would enhance the educational value of the manuscript.

In response to your feedback, we have made the following revisions:

  1. Expanded discussion on congenital anomalies: We have added a more detailed section on congenital uterine anomalies, including their role in the development of IUAs. This section covers various types of congenital malformations (e.g., septate uterus, Robert`s uterus) and their association with an increased risk of intrauterine adhesions. This will provide a more comprehensive view for readers.
  2. Clarification regarding the D&C as a causative factor: We acknowledge your concern regarding the low number of D&C procedures identified as a causative factor in our study (only 2 cases). This is a valid point, and we have provided further clarification in the manuscript. Endometrial biopsy via D&C is quite common in our country. However, the low number of cases in the two maternities from which the cases were selected is due to the younger age of the patients included in the study, where the prevalence of D&C for biopsy is lower, particularly considering the lower prevalence of oncological diseases in this age group. Furthermore, other benign uterine pathologies, such as polyps and fibroids, are typically diagnosed via ultrasound prior to treatment, with hysteroscopic procedures being the preferred method of management. While hysteroscopic surgery (Hysc) is more commonly performed and may be more frequently linked to the formation of intrauterine adhesions (IUAs), D&C procedures were conducted in very specific contexts during the study period. Furthermore, regional variations in the frequency and indications for D&C versus hysteroscopic procedures may exist. In Romania, hysteroscopy may have become the more commonly employed first-line intervention for uterine conditions, which could account for the lower incidence of D&C cases observed in our study. We have updated the manuscript to include this contextual explanation for enhanced clarity.

Additionally, the selection of patients for the study involved those diagnosed preoperatively and intraoperatively with intrauterine adhesions, focusing on risk factors, rather than patients with risk factors from which the incidence of intrauterine adhesions was analysed. We hope these revisions address your concerns and contribute to a clearer understanding of the study's findings. Thank you again for your constructive feedback, which has helped improve the manuscript.

Comments on the Quality of English Language

Authors made small errors in writing and some tables still have Roumanian names. Change them.

Thank you for your feedback. We will correct the minor writing errors and ensure that all tables are updated with the appropriate English names. We will revise the manuscript accordingly and make sure that all terms, especially those in Romanian, are replaced with their English equivalents for clarity and consistency.

Round 2

Reviewer 1 Report

Comments and Suggestions for Authors

The authors' corrections have significantly improved the manuscript. I have no further comments.

Author Response

Comments and Suggestions for Authors: The authors' corrections have significantly improved the manuscript. I have no further comments.

Response: Thank you so much for your kind feedback. I greatly appreciate your thoughtful review and am pleased to hear that the revisions have met your expectations. Your insights were invaluable in improving the manuscript, and I’m grateful for the time you took to review the work.